# Intrinsically Disordered Proteins as Regulators of Transient Biological Processes and as Untapped Drug Targets

**DOI:** 10.3390/molecules26082118

**Published:** 2021-04-07

**Authors:** Yusuke Hosoya, Junko Ohkanda

**Affiliations:** Academic Assembly, Institute of Agriculture, Shinshu University, 8304 Minami-Minowa, Kami-Ina, Nagano 399-4598, Japan; 19as111g@shinshu-u.ac.jp

**Keywords:** intrinsically disordered proteins, protein–protein interactions, liquid–liquid phase separation, drug discovery

## Abstract

Intrinsically disordered proteins (IDPs) are critical players in the dynamic control of diverse cellular processes, and provide potential new drug targets because their dysregulation is closely related to many diseases. This review focuses on several medicinal studies that have identified low-molecular-weight inhibitors of IDPs. In addition, clinically relevant liquid–liquid phase separations—which critically involve both intermolecular interactions between IDPs and their posttranslational modification—are analyzed to understand the potential of IDPs as new drug targets.

## 1. Introduction

Approximately 20,687 protein-coding genes have been identified [1] in the human genome, and 311,962 protein–protein interactions (PPIs) have been predicted [2]. The proteome consists of more than half a million proteins, due to the epigenomic regulation of nucleic acids [3,4], and protein post-translational modification (PTM) [5,6,7].

The terms “intrinsically disordered proteins” (IDPs) and “intrinsically disordered regions” (IDRs) (here generally referred to as IDPs) were introduced over a decade ago to describe the diversity of flexible proteins that are unfolded under physiological conditions and lack well-defined tertiary structures [8]. IDPs and IDRs comprise a flexible stretch of polar residues, and are predicted to be present in about 40% of all human proteins [9]. From a large number of nearly isoenergetic conformations, these proteins fold into their most stable conformation under optimal conditions (e.g., temperature, solvent composition, pH) [10,11]. This structural flexibility enables their molecular recognition of PPIs and PTMs, allowing for the regulation of complex biological processes, such as chromatin organization, cell cycle progression, proliferation, translational regulation, immune response, autophagy, and synaptic formation [9,11]. Studies have revealed that the dysregulation of such processes is linked to various diseases, such as cancers and neurodegenerative diseases, highlighting the potential of IDPs to be a new class of drug targets [12,13,14,15,16,17,18].

Disordered flexible proteins and regions are involved in liquid–liquid phase separation (LLPS), a process that generates non-membrane liquid-like cellular compartments [19]. These transiently generated non-membrane organelles regulate the intracellular and intranuclear localization of compounds, triggering a series of biological reactions in a time- and space-specific manner [20,21,22]. The flexible regions of proteins undergo PTM during LLPS, providing a major driving force for the transient formation and dissociation of molecular assemblies in cells [23,24,25]. Dysregulation of LLPS localization is implicated in neurological diseases, such as Alzheimer’s disease and amyotrophic lateral sclerosis (ALS) [26,27,28]. Understanding the molecular basis of the dynamics of LLPS could thus provide a new perspective on chemical approaches for regulating cellular processes. 

The conformations of IDPs are mostly transient and lack a fixed, three-dimensional structure, making it extremely difficult to develop inhibitors based on the conventional structure–function paradigm [29]. There has been a huge effort over the past several decades to analyze and predict the dynamic conformational changes of IDPs using various spectroscopic and computational approaches, and these biophysical studies have substantially advanced our understanding of the molecular mechanisms underlying transient biological events. This review discusses examples of clinically relevant IDPs and the low-molecular-weight inhibitors of these proteins, highlights recent studies elucidating the biological functions and molecular mechanism of LLPS, and discusses the potential of IDPs as future drug targets. 

## 2. Inhibitors of Intrinsically Disordered Proteins

A number of clinically relevant IDPs have been identified, and their structural modulation and the inhibition of their PTMs by synthetic molecules are promising therapeutic approaches. In this review, efforts aimed at drug discovery for several examples of IDPs that are implicated in cancers and neurodegenerative diseases are highlighted.

### 2.1. Intrinsically Disordered Proteins (IDPs)

In 2000, Dunker et al. proposed the concept of IDPs, which are proteins that form an ensemble of multiple conformations under physiological conditions [30]. From a large number of nearly isoenergetic conformations, these proteins fold into their most stable conformation through interactions with other biomolecules and/or through their PTMs [10,31]. IDPs are present in 6% to 33% of bacterial proteins, 9% to 37% of archaeal proteins [32], and 35% to 50% of eukaryotic proteins [12,32]. IDPs are composed of disordered stretches, containing about 40 amino acid residues [9]. Analysis of these sequences showed that they frequently contain a high percentage of Ser, Gly, Pro, Asn, and Gln, and a low frequency of hydrophobic amino acids [33,34,35,36,37,38]. The sequences also contain tandem repeats of these amino acids, as well as charge-compensating Lys, Arg, Glu, and Asp [33]. The exposed and flexible amino acid residues are often targeted for PTMs, such as phosphorylation [39], acetylation [40], methylation [41], and small, ubiquitin-like modification (SUMO) [42]. The dynamic conformational changes of IDPs are also greatly influenced by pH [43] and temperature [44]. Computational algorithms, such as Disprot [45], CDF [46], FoldIndex [47], TopIDP [48], and Ucon [49], can simulate the degree of disorder of a protein. These tools have enabled the identification of a number of IDRs in proteins involved in intracellular signal transductions and tumorigenesis, such as p53, c-Myc/Max, CBP/p300, and hypoxia-inducible factor 1 (HIF-1) [50,51,52]. Early biophysical studies of IDPs characterized their surface charges and hydrophilicities using small-angle X-ray scattering [53], single molecule Förster resonance energy transfer [54], nuclear magnetic resonance (NMR) [55], and surface plasmon resonance [56]. Enormous effort over the past decade has been focused on exploring low-molecular-weight compounds that regulate IDPs in vitro and in vivo, and examples of these studies are discussed in the next section. 

### 2.2. c-Myc and Max

The Myc protein is a transcription factor that binds to target DNA by forming a heterodimer with the Max protein; this heterodimer can activate or repress transcription [57]. The target genes are diverse, and affect cell growth, metabolism, apoptosis, and extracellular matrix formation [58,59]. The Myc protein is often overexpressed in human cancer cells, and it inhibits differentiation, induces genomic instability, and promotes angiogenesis [60,61]. This protein, made of 439 amino acids, comprises an *N*-terminal transactivation domain (TAD), a transcriptional regulatory Myc box (MB), and a *C*-terminal basic helix–loop–helix–leucine zipper (bHLHZ) domain [62]. The TAD and bHLHZ domains are largely disordered [63]. Residues 98–111 of the TAD bind to TATA-binding proteins [62], and residues 128–142 in the MB interact with multiple transcriptional regulators, including histone acetyltransferase [62,63].

Prochownik et al. adopted a yeast two-hybrid-based approach for screening a chemical library to identify inhibitors of the Myc–Max interaction [64]. The bHLHZ domains of the Myc and Max fused to the DNA-binding domain linked to the TAD of the yeast. The Gal4 transcription factor was used to screen a chemical library of 10,000 low-molecular-weight compounds; seven compounds—including **1** (Table 1)—that inhibit the induction of β-galactosidase were identified. These compounds were shown to inhibit the proliferation of Rat1a cells and reduce tumor growth significantly in nude mice [64]. 

Schultz et al. carried out a protein-fragment complementary assay, exploiting full-length Max fused to the bHLHZ domain of Myc to split Gaussia luciferase Gluc1 and a Gluc2 [65]. A chemical library of about 400,000 drug-like molecules was screened, and sAJM589 (Table 1) was found to exhibit significant inhibition activity against luciferase luminescence, with an IC_50_ = 1.8 μM [65].

### 2.3. P27

The protein p27 belongs to the family of KIP1 inhibitors of cyclin-dependent kinases (CDKs) [73,74,75]. The *C*-terminal domain contains a nuclear localization signaling domain, and multiple phosphorylation sites required for recruitment to DNA. The CDK2 phosphorylates the *C*-terminal Thr187 of p27, promoting polyubiquitination and degradation by SCFSkp2 ubiquitin ligase [76,77]. In breast cancer, Thr187 is inappropriately phosphorylated, promoting cell migration [78,79,80]. 

The N-terminal kinase inhibitory domain (KID) of p27, comprising D1, D2, and LH subdomains, is disordered, and folds through interactions with CDK2 and cyclin A [81]. Using a model protein containing only the p27 KID, one-dimensional ^1^H WaterLOGSY and STD NMR methods were used to screen a chemical library of 1100 compounds from the Ro3 collection and an in-house library of 1222 compounds [66]. Nine molecules were found to inhibit the interaction between p27 and the CDKs, of which SJ403 (Table 1) showed an IC_50_ of 475 μM and restored the activity of CDK2/cyclin A in vitro [66]. The molecular dynamics simulations and NMR analysis suggested that SJ403 specifically binds to aromatic residues, implying that designing new compounds that target these residues could exhibit improved affinity. 

### 2.4. Ewing’s Sarcoma-Friend Leukemia Integration 1 (EWS-FLI1)

Ewing’s sarcoma (EWS) is a bone and soft tissue carcinoma that produces an Ewing’s sarcoma–Friend leukemia integration 1 (EWS-FLI1) fusion protein, which is a transcription factor that may be a drug target for the disease [82,83]. A number of oncogenic genes are related to EWS-FLI1, such as IGFBP3, GSTM4, CDKN1A, TGFBRII, VEGF, CAV1, E2F8, FOXO1, and NFKBIL2 [84]. An EWS-FLI1 consists of 476 amino acid residues, and has a highly disordered *N*-terminal region containing the EWS-derived transcriptional activation domain (EAD; 264 residues), whereas the *C*-terminal region may be somewhat less disordered [85,86]. The binding of EWS-FLI1 to RNA helicase A is important for its oncogenic function. Library screening of 3000 compounds, using surface plasmon resonance, identified the small compound NSC635437 (Table 1), which binds to EWS-FL1 [67]. The structurally similar YK-4-279 (Table 1) was shown to bind to EWS-FL1, with a *K*_d_ value of 9 µM, and inhibit the interaction of EWS-FL1 and RNA helicase A. The YK-4-279 exhibited cytotoxicity against the Ewing sarcoma family of tumors cells (ESFT cells), with an IC_50_ value of 0.5–2 μM, and was inactive toward cells lacking EWS-FL1. Administration of 60–75 mg/kg of YK-4-279 reduced the growth of ESFT orthotopic xenografts [67]. These results support the potential versatility of the chemical disruption of clinically relevant IDPs. 

### 2.5. Nuclear Protein 1 (NUPR1)

Nuclear protein 1 (NUPR1) is a disordered transcription regulator that converts stress signals into various gene expressions related to cellular stress response [87]. NUPR1 is overexpressed in pancreatic and breast cancers, and is involved in several tumorigenic cellular processes, such as cell cycle regulation [88], apoptosis [89], senescence [88], cell wetting and migration [90,91], and DNA repair [92]. Human NUPR1 is composed of 82 amino acid residues, with an *N*-terminal PEST (Pro/Glu/Ser/Thr-rich) region and a positively charged *C*-terminal nuclear target signal region.

Using a recombinant *N*-terminal His-tagged NUPR1, Jose et al. screened a chemical library of FDA-approved compounds using fluorescent thermal denaturation, and the antipsychotic Trifluoperazine (Table 1) was identified as a binder to NUPR1 [68]. An in silico study led to the design of a modified analog of Trifluoperazine: ZZW-115 (Table 1). ITC experiments showed that the ZZW-115 binds to NUPR1, with a *K*_d_ of 2.1 μM [69]. The ZZW-115 exhibited cytotoxicity against various cell lines related to pancreatic ductal adenocarcinoma, with low micromolar IC_50_ values, and was found to be effective against drug-resistant pancreatic cells (MiaPaCa-2). The ZZW-115 that was administered to xenograft mice (5 mg/kg/day) for 30 days remarkably decreased tumor size, and eliminated the tumor by inducing necrosis [69]. 

### 2.6. Hypoxia-Inducible Factor-2α (HIF-2α)

Hypoxia-inducible factors (HIFs) are a group of hypoxic environment inducers that enhance the expression of hundreds of downstream genes, and are closely related to cancer progression [93,94]. HIF proteins consist of DNA-binding and dimerization domains, a basic helix–loop–helix (bHLH) domain, Per-ARNT-Sim-A and B (PAS-A and PAS-B) domains, an oxygen-dependent degradation domain (ODD), and a transactivation domain. Under normoxic conditions, the HIF-prolyl hydroxylases the hydroxylate proline residues in the ODD region. The von Hippel–Lindau tumor suppressor protein (VHL), a component of the E3 ubiquitin ligase complex, is then recruited, promoting the rapid degradation of the HIF proteins. In contrast, under hypoxic conditions, HIF proteins are stabilized, accumulate in the nucleus, and form heterodimers with the aryl hydrocarbon receptor nuclear translocator (ARNT), resulting in the overexpression of hypoxia response elements involved in cell growth, angiogenesis, glucose metabolism, pH regulation, and cell survival/apoptosis [95]. The PAS-B regions of the HIF and ARNT are critical for heterodimer formation [96]. 

Bruick et al. demonstrated that the crystal structures of the PAS-B domains of HIF-2α and ARNT are almost identical, and form a 290-Å-long hydrophobic cavity [70]. HSQC NMR library screening identified THS-044 (Table 1), which binds to the hydrophobic cavity with low micromolar affinity. This THS-044 binding reduced the affinity of HIF2α to ARNT by approximately one-third. The crystal structures of the ternary complex of PAS-B and HIF2α bound to THS-044, and of ARNT bound to THS-044, revealed that the compound disordered the Met252 and His293 side chains in the HIF-2α hydrophobic cavity [70]. 

### 2.7. BMAL1 and CLOCK

The dysregulation of circadian rhythms causes abnormal expression levels of downstream proteins, leading to aging [97,98], metabolic diseases [99], insomnia and tumorigenesis [100]. Small molecules capable of regulating a series of circadian transcription factors would thus be useful for developing therapeutic agents for various chronic diseases. The circadian transcription activators—brain and muscle ARNT-like 1 (BMAL1) [101] and circadian locomotor output cycles kaput (CLOCK) [102]—are flexible proteins. More than 30% of BMAL1 proteins, and nearly 60% of CLOCK proteins, are predicted to be disordered [103,104]. The BMAL1 and CLOCK proteins form a heterodimer that binds to E-box regulatory elements in the *Period* (*Per1* and *Per2*) and *Cryptochrome* (*Cry1* and *Cry2*) genes, and activates their transcription during the daytime. At night, the protein products *PER* and *CRY* accumulate, dimerize, translocate to the nucleus, and bind to the BMAL1/CLOCK heterodimer, which reduces its transcriptional activity [105]. This transcriptional feedback loop is central to maintaining the 24 h circadian clock in mammals.

Kavakli et al. virtually screened approximately two million small molecules to identify compounds that specifically bind to the dimerization interface of CLOCK [71], and identified CLK8 (Table 1). A cell-based reporter assay showed that CLK8 enhances the amplitude of the circadian rhythm. Docking simulations of CLK8 and CLOCK suggested that CLK8 binds to Phe-80 and Lys-220 in the cavity located between the bHLH and PAS-A of CLOCK. A co-immunoprecipitation assay and a cell-based study demonstrated that CLK8 disrupts the interaction between CLOCK and BMAL1, as well as the translocation of CLOCK into the nucleus of U2OS cells. Animal studies further indicated that CLK8 inhibits the interaction between CLOCK and BMAL1 by specific binding to CLOCK, and interferes with the nuclear translocation of CLOCK in vitro and in vivo [71]. 

We have designed an in vitro screening system based on the fluorescence polarization (FP) change that occurs upon the binding of recombinant BMAL1 and CLOCK to a fluorogenic *Per2* E-box DNA fragment [72]. A chemical library of almost 2000 low-molecular-weight compounds was screened, and we identified 5,8-quinoxalinedione **2** (Table 1), which significantly inhibits DNA-binding at low micromolar concentrations. A structure–activity relationship study, followed by a series of biochemical analyses, including a cysteine capping experiment, revealed that **2** likely reacts covalently with the PAS region of BMAL1 and inhibits dimerization, disrupting BMAL1 from binding to DNA. These results suggest that covalent reagents may provide a molecular basis for the development of IDP inhibitors.

## 3. Liquid–Liquid Phase Separation

### 3.1. Liquid–Liquid Phase Separations

IDPs are implicated in the regulation of LLPS in cells, and in the localization of biomolecules [106]. LLPS is driven by intermolecular interactions, such as charge–charge, charge–π, and π–π stacking between the amino acid residues of proteins [19,107,108,109,110,111]. The repetition of short motifs, such as Tyr Gly-/Ser-, Phe Gly-, Arg Gly-, Gly Tyr-, Lys Ser Pro Glu Ala-, Ser Tyr-, and Gln/Asn-rich regions, provides the hydrophobicity, polarity, and charge required to drive LLPS [112]. 

LLPS drives the formation of membrane-free cell organelles, such as nuclear organelles [113,114,115]. These organelles include promyelocytic leukemia (PML) bodies, the nucleolus, and stress granules containing numerous proteins and RNA molecules [112,116,117]. LLPS is reversible and dynamic; thus, non-membrane organelles and other condensates alternate between formation and dissociation [118,119]. The equilibrium between the formation and dissociation of these biomolecular assemblies is important for maintaining intracellular homeostasis [106,120]. In contrast, if the equilibrium of molecular localization is disrupted, biological mechanisms can then be disrupted, and abnormalities in the dynamics of molecular condensates, such as nucleoli and stress granules, cause tumorigenesis and neurological disease [119,121,122]. 

The nucleolus is the largest membrane-free structure in the eukaryotic nucleus [117]. The nucleolus is involved in the synthesis of ribosomes, and the expression of ribosome-related nucleic acids, and the number and activity of functional ribosomes, are finely regulated [115,117]. This regulation is controlled by the localization of substances through LLPS [115,117]. Abnormalities in LLPS can disrupt the balance between rRNA, rDNA and ribosome production, and an excess of functional ribosomes can lead to excessive translation, and cause tumorigenesis [118,123]. 

Stress granules (SGs) are ribonucleoprotein (RNP) particles that assemble in response to environmental stress [122,124]. SGs contain defective mRNAs, defective ribosomal products (DRIPs), RNA-binding proteins (RBPs), and molecular chaperones [43], temporarily sequestering translationally deficient peptides and ribosomes that increase under stress conditions [43], and guiding them toward repair or degradation [43]. SGs are highly dynamic structures, and are degraded within a few hours by the ubiquitin–proteasome system (UPS) or autophagy [125,126]. However, excessive accumulation of misfolded proteins and DRIPs interferes with SG dynamics and leads to SG aggregation [125,127], which, in turn, disrupts intracellular proteostasis and ribostasis, leading to neurodegeneration [125,127,128]. 

RBPs and nucleic acids are considered to be scaffolds that may be important in the formation and dissociation of intracellular granules [129]. Some RBPs are rich in IDRs, and their post-translational modification regulates the formation and dissociation of aggregates [119,122]. Synthetic molecules capable of regulating the dynamics of these aggregates could provide a new approach to the development of therapeutic agents for cancer and neurological diseases. The following are examples of disease-relevant IDPs implicated in PTM-driven LLPS, and may serve as potential drug targets.

### 3.2. Tau

Tau proteins are soluble, neuron-specific microtubule-binding proteins, and primarily regulate microtubule stability by interacting with tubulin and recruiting signaling proteins [130]. Tau is also a major component of neurofibrillary tangles in Alzheimer’s disease and frontotemporal dementia [130]. The longest human Tau isoform is 441 amino acids, and consists of two *N*-terminal repeats (N1, N2), two proline-rich repeats (P1, P2), four pseudo-repeats (R1-R4), and another proline-rich repeat (P3) [131]. 

Tau441 is a largely disordered protein, with 22 serine/threonine phosphorylation sites [132], of which 15 are located in the disordered regions [130]. Experiments with recombinant, full-length Tau441 produced by bacteria demonstrated that LLPS was triggered primarily by electrostatic intermolecular interactions, and did not require phosphorylation [132]. However, another in vitro study showed that the phosphorylation of Tau441 was required to initiate LLPS [130]. LLPS of phosphorylated tau was shown to be dependent on hydrophobic interactions, whereas LLPS by non-phosphorylated tau was dependent on ionic interactions [130,132]. Therefore, the driving force of LLPS by tau likely differs depending on its phosphorylation state. 

### 3.3. DEAD-Box Helicase 3 X-Linked DDX3X

DEAD-Box Helicase 3 X-Linked (DDX3X) is 662 amino acid residues long, consists of two domains (helicase ATP-binding (211-403) and helicase *C*-terminal (414-575)), and has IDR regions at the *C*- and *N*-termini [133,134]. DDX3X is a member of the DEAD-box family of RNA helicases, and is involved in double-stranded RNA unwinding and pre-mRNA splicing [135]. 

Analysis of an HDAC6-dependent acetylome data set identified several acetylated lysines in the *N*-terminal IDR that were involved in the incorporation of DDX3X into SGs [135]. To understand the relationship between DDX3X acetylation and SG-uptake, acetyl-mimetic (K→Q) mutants and acetyl-dead (K→R) mutants were expressed in DDX3X knockout cell lines [135]. The expression of acetyl-dead DDX3X increased SG levels, whereas the expression of the acetyl-mimetic mutant decreased SG levels [135]. Unacetylated DDX3X interacted with a large number of SG components, whereas acetylmimetic DDX3X lost the ability to interact with SG components [135]. This study suggested that acetylation and deacetylation of the lysines in DDX3X spatiotemporally regulates LLPS formation, and thus membrane-less organelle formation, and proposed a potential therapeutic rationale for targeting histone deacetylases and histone acetyltransferases in neurodegenerative diseases [135]. 

### 3.4. Fused in Sarcoma

Fused in sarcoma (FUS) is a frequently studied protein, because its phase separation in vivo has been linked to ALS [136]. FUS is 526 amino acids long [137] and comprises a QGSY-rich region, a Gly region, an RNA recognition motif, an RGG domain, and a ZNF domain [137]. FUS is involved in cell proliferation, DNA repair, transcriptional regulation, and mRNA splicing regulation [138]. The function of FUS depends on maintenance of the aggregation–dispersion loop of LLPS. When LLPS is out of balance, FUS aggregates and causes neurological diseases [136]. 

The QGSY-rich region of the FUS is a highly hydrophobic IDR region, called the prion-like domain (PrLD) [121]. The PrLD contains 32 phosphorylation sites, of which 12 have been identified as phosphatidylinositol 3-kinase-related kinase (PIKK) family kinase consensus sites [138]. Phosphorylation substitutions (S/T→E) at 6 or 12 PIKK consensus sites reduce the ability for FUS to undergo phase separation and fibril aggregation [139]. A decrease in cytoplasmic aggregation is also observed with an increase in phosphomimetic substitutions [139], suggesting that the FUS is a potential therapeutic target for inhibiting pathological aggregate formation. When the arginine-rich RGG domain of FUS is citrullinated, the localization of FUS to the stress granules is reduced [140]. This removal of the arginine side chain charge inhibits the cation–π interaction between the arginine and tyrosine residues, resulting in phase separation [141]. Molecules that modify arginine side chains could inhibit the formation of FUS-LLPS and regulate the aggregation process [141].

### 3.5. Transactive Response DNA-Binding Protein 43 kDa (TDP-43)

The transactive response DNA-binding protein 43 kDa (TDP-43) is a 414-residue protein belonging to the heterogeneous nuclear ribonucleoprotein (hnRNP) family [142]. TDP-43 plays a role in mRNA metabolism, localization, and transport [137], and consists of a structured *N*-terminal domain, two RNA recognition motifs (RRM1 and RRM2), and a highly unstable IDR at the *C*-terminus [142]. The *C*-terminal domain interacts with hnRNPs and the FUS [142]. The reversible localization and dissociation of TDP-43 by LLPS induces local RNA translation and splicing changes in response to neuronal stimuli [137,143,144]. However, the disruption of LLPS homeostasis by mutation causes a loss of splicing function and fibril formation [145]. 

The PrLD at the *C*-terminus of TDP-43 is hyperphosphorylated and aggregated in ALS [144]. Phosphomimetic substitutions (S→D) of serines 409 and 410 in this domain reduce LLPS [146]. Moreover, phosphorylation at serine 48 in the structured *N*-terminal domain is implicated in the regulation of the LLPS [146]. These results indicate that not only the flexible *C*-terminal domain, but also the PTM of the structured *N*-terminal, are involved in the regulation of TDP-34-LLPS, suggesting that the structure-based design of inhibitors targeting the *N*-terminal of the protein may be possible. 

## 4. Summary and Future Perspectives

Disordered and flexible proteins clearly play central roles in the control of transient biological functions implicated in diverse cellular responses. A number of clinically relevant IDPs have been identified, and their structural modulation and the inhibition of their PTMs by synthetic molecules are promising therapeutic approaches. Intensive medicinal efforts have begun to yield potent inhibitors, but further experimental techniques for generating focused libraries, as well as evaluation methodologies, must be developed. To this end, the exploration of new chemical space, not limited to drug-like small molecules, may help advance the IDP-targeting pharmaceuticals. For example, multivalent agents that have been widely studied for the regulation of intracellular PPIs could provide potential chemical platforms for IDPs [147]. Examples include semi-synthetic analogs of natural products, cell-permeable peptides, and peptide-conjugates. In addition, the application of covalent drugs may also serve as a promising approach for IDPs. 

Informative technologies, such as molecular dynamics simulations, have developed rapidly in recent years, and are effective for analyzing the functions of IDPs and for predicting compounds that interact with IDPs. Studies have suggested that the combination of computer-aided approaches and experimental validations targeting IDPs is essential for drug discovery. Compounds that induce changes in the orientation of amino acid residues surrounding the binding site, or that form strong bonds, such as covalent bonds, may represent promising candidates for IDP modulators. In vitro evaluation systems, using appropriately truncated recombinant IDPs, would provide robust platforms useful for high-throughput screening.

Advances in understanding the molecular mechanism of LLPS assembly now allow for control of the formation and dissociation of LLPS by synthetic molecules, and these chemical approaches will expand to include clinically relevant, non-membrane organelles. To this end, appropriate evaluation platforms for identifying and evaluating synthetic organic molecules, as well as animal models, must be developed, along with strategies allowing for the spatial–temporal control of LLPS in cells and animals. The exploration of bioorganic approaches for controlling these untapped dynamic protein assemblies is just beginning, and many exciting outcomes are anticipated. 

## Figures and Tables

**Table 1 molecules-26-02118-t001:** Examples of low-molecular-weight inhibitors of intrinsically disordered proteins (IDPs).

Compound	Structure	Target	Method ^1^	Activityin Vitro	Activity in Cells	Refs
**1**	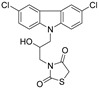	c-Myc	YTH ^2^	21 μM	15.1 μM	[64]
sAJM589	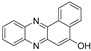	c-Myc	PCA ^3^	1.8 μM ^7^	1.2 μM	[65]
sJ403	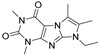	p27	NMR	2.2 mM ^8^	n/a	[66]
NSC635437	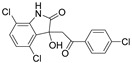	EWS-FLI1	SPR ^4^	n/a	20 μM	[67]
YK-4-279	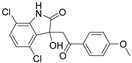	EWS-FLI1	SPR ^4^	9.48 μM ^8^	0.9 μM	[67]
Trifluoperazine	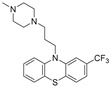	NUPR1	TSA ^5^	5.2 μM ^8^	26 % at 10 μM	[68]
ZZW-115	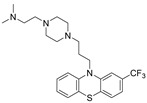	NUPR1	in silico	2.1 μM ^8^	1.03 μM	[69]
THS-044	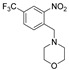	HIF-2α	NMR	2 μM ^8^	n/a	[70]
CLK8	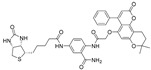	CLOCK	in silico	n/a	20 μM	[71]
**2**	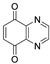	BMAL1	FP ^6^	~1 μM ^7^	n/a	[72]

^1^ Method for screening. ^2^ Yeast two-hybrid. ^3^ Protein-fragment complementation assay. ^4^ Surface plasmon resonance. ^5^ Thermal shift assay. ^6^ Fluorescence polarization. ^7^ IC_50_ value. ^8^
*K*_d_ value.

## Data Availability

Not applicable.

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
