# Peer review of "Intrinsically Disordered Proteins as Regulators of Transient Biological Processes and as Untapped Drug Targets"

_molecules, 2021, doi:10.3390/molecules26082118_

Round 1
Reviewer 1 Report
The manuscript by Hosoya et al. is a review presenting intrinsically disordered proteins (IDPs) and is focused on their growing interest as a novel type of promising drug targets. The introduction presents IDPs as well as their central role in non-membrane liquid-like cellular compartments via their capacity to induce liquid-liquid phase separation (LLPS). The two main parts describe inhibitors of biologically relevant IDPs and of LLPS respectively with sub-sections organized around defined biological targets.
The review is dense but relatively clear and well-written. Illustrations are minimal (only one figure). Citations are appropriate and numerous. My main concern and comment would be that the review, which in its current state appears essentially as a long list of recently developed inhibitors against IDPs, would benefit from a section discussing in more details the perspective of that growing field (challenge, new methodologies...). The authors started this discussion in the summary but it is too short and a larger impact would be achieved with a more elaborated section dedicated to future directions.
Author Response
Response to Reviewer 1:
We appreciate the constructive suggestion by this reviewer. Several suggestions for the chemical exploration targeting IDPs in the future (i.e. application of multivalent molecules and covalent inhibitors) have been added in the Summary and Future Perspectives.
Reviewer 2 Report
The review is timely, well written and highlights the growing relevance of intrinsically disordered regions, especially within the context of the drug discovery. The section on LLPS is also insightful.
Section 2 can be slightly improved:
How were the proteins selected for analysis and are there any similarities between them? A few transitions sentences at the end of section 2-1 would help the review.
It would have also been easier to put the structures of Figure 1 into a Table, with the protein target and IC50, activity, other parameters. Also, a cartoon-schematic of each protein structure would be helpful to see where the IDR is within the protein.
Finally, are there any drugs that target structured regions of these proteins, and how do they compare to IDR-targeting drugs.
Also, The first paragraph states, “500,000 proteins due to the epigenomic regulation” but also lists “ 311,962 protein–protein interactions” have been identified, which is less than 1:1 protein-protein interactions.
Author Response
Response to Reviewer 2:
We appreciate all the positive suggestions from this reviewer. Belief introduction has been added before 2-1.
We transformed Figure 1 into Table 1, according to the suggestion by this Reviewer. Thank you for the suggestion.
>Finally, are there any drugs that target structured regions of these proteins, and how do they compare to IDR-targeting drugs.
For the IDP proteins discussed in this review, no example of drugs that bind to their structural regions have been reported to my best knowledge. However, there are examples of drugs that specifically bind to the structural regions of IDP-containing proteins, such as enzymes. For example, K-Ras protein possesses C-terminal IDR and several inhibitors identified from library screenings are small compounds that bind to a hidden pocket in the structured region. These compounds have been shown to block K-Ras' GTPase activity, but not its posttranslational lipid modifications. Thus, the K-Ras IDR inhibitors would work in a different mechanism of action and may advantageous for overcoming drug resistance.
>Also, The first paragraph states, “500,000 proteins due to the epigenomic regulation” but also lists “ 311,962 protein–protein interactions” have been identified, which is less than 1:1 protein-protein interactions.
This sentence means that, although human genes code approximately 20 thousand proteins, actual number of working proteins is estimated as more than half a million due to the epigenomic regulation. This implies that enormous number of unknown (not predicted yet) PPIs should exist, and they are likely involving IDRs and IDPs.
Reviewer 3 Report
The review entitled “Intrinsically Disordered Proteins as Regulators of Transient Biological Processes and as Untapped Drug Targets" by Junko Ohkanda and Yusuke Hosoya is clear and informative. The manuscript is well written and easy to understand. Therefore, I think the work fits well inside the goals of “Molecules” and its publication could be appropriate.
Author Response
We appreciate this reviewers' supportive comments.
Round 2
Reviewer 1 Report
I am satisfied with the new version even though a more extended discussion on the perspective would have appreciated.